# The Joint Solvation Interaction

**DOI:** 10.3390/e26090749

**Published:** 2024-09-01

**Authors:** Ali Hassanali, Colin K. Egan

**Affiliations:** The “Abdus Salam” International Centre for Theoretical Physics, I-34151 Trieste, Italy

**Keywords:** solvation thermodynamics, hydrophobic effect, hydrophobic interaction, hydrophilic interaction, collective effects, cluster expansion

## Abstract

The solvent-induced interactions (SIIs) between flexible solutes can be separated into two distinct components: the solvation-induced conformational effect and the joint solvation interaction (JSI). The JSI quantifies the thermodynamic effect of the solvent simultaneously accommodating the solutes, generalizing the typical notion of the hydrophobic interaction. We present a formal definition of the JSI within the framework of the mixture expansion, demonstrate that this definition is equivalent to the SII between rigid solutes, and propose a method, partially connected molecular dynamics, which allows one to compute the interaction with existing free energy algorithms. We also compare the JSI to the more natural generalization of the hydrophobic interaction, the indirect solvent-mediated interaction, and argue that JSI is a more useful quantity for studying solute binding thermodynamics. Direct calculation of the JSI may prove useful in developing our understanding of solvent effects in self-assembly, protein aggregation, and protein folding, for which the isolation of the JSI from the conformational component of the SII becomes important due to the intra-species flexibility.

## 1. Introduction

Many important physical processes in solution are influenced by solvent-induced interactions (SIIs), for example aggregation [1,2,3,4], self-assembly [5], and biochemical processes [6,7], in particular protein folding [8,9,10,11,12]. The hydrophobic interaction [13,14,15,16] (the main driving force underlying the hydrophobic effect) is the most widely known example of an SII [17], but the general phenomenon of indirect interactions between two or more solutes mediated through the surrounding solvent molecules is ubiquitous. Many of the theoretical studies on SII consider the indirect interaction between effectively rigid solutes, with the classic examples being the hydrophobic interaction between two hard spheres or two methane molecules in water. However, this limitation obscures the fact that SIIs include two distinct effects: the effect of solvation on the conformation of each solute (which in turn affects how the solutes interact with each other) and the effect due to the solvent accommodating the solutes simultaneously, the latter of which we refer to as the *joint solvation interaction* (JSI). The JSI is typically the only SII considered since rigid solutes have no solvent-induced conformational changes.

The solvation of a single solute induces a change in the solvent structure and dynamics (in particular due to solute–solvent interactions and correlations). For example, excluded volume effects between the solute and solvent lead to cavity formation. In the case of aqueous solvation, cavity formation may disrupt the surrounding hydrogen bonding network [3]. Additionally, electrostatic solute–solvent interactions may lead to orientational polarization of the solvent [18]. All of these effects are accompanied by changes in thermodynamic state functions (energy, entropy, etc.), which determine the corresponding free energy of solvation.

When two or more solutes are solvated independently, the total free energy change is the sum of the individual free energies. In particular, if two solutes are positioned far enough away from each other, they will each lie outside the correlation length of the solvation structure of the other solute. This is the prototypical example of what we call *disjoint solvation*. As the two solutes approach one another, there will generally be a nonlinear interference between the two solvation structures, leading to a nonadditive total solvation free energy due to their (simultaneous) *joint solvation*.

For example, while disjointly solvated solutes may each have *n* nearest solvent neighbors on average, when the two solutes come into contact, excluded volume effects may require that one or more solvent neighbors are expelled from the region between the two solutes, as depicted in Figure 1. As a result, there may be a net change in the entropy, due to the solute–solvent structuring, and a change in the solute–solvent energy, $E_{U,V}$, going from $E_{U,V}^{\mathrm{disj}.}∼2n\epsilon$ for disjoint solvation to $E_{U,V}^{\mathrm{joint}}∼2\left(n-1\right)\epsilon$ at contact (for average pairwise solute–solvent interaction energy, ϵ). These kinds of indirect, solvent-mediated contributions to the total potential of mean force between the solutes represent an important class of collective effects in solvation thermodynamics, and are thus suited to analysis within the framework of the mixture expansion (ME) [19].

The ME, a coarse-grained cluster expansion, provides a general framework for free energy decompositions. While formally grounded in the theory of the microscopic cluster expansion (CE) [20,21,22,23,24,25,26], the objectives of the two theories are distinct. While the CE requires the direct evaluation of integral equations, the ME leverages free energy calculations from full-dimensional molecular dynamics (MD) simulations, broadening the range of accessible problems. The ME is an analysis tool rather than a computational tool, allowing one to isolate collective many-body thermodynamic effects. In our previous work [19], we used the ME to decompose interfacial binding free energies into contributions due to specific collective effects in order to interpret and understand (surface-sensitive) sum frequency generation experimental measurements [27].

Here, we generalize the ME by way of the interaction expansion (see Section 3), allowing for a more detailed free energy decomposition scheme. In particular, the flexibility of this generalized ME provides a framework in which the JSI can be extracted from the total SII. The key idea is that the use of partially connected couplings allows one to construct a system in which two flexible solutes can be disjointly solvated while still interacting with each other. Subtracting the disjointly solvated binding free energy from the fully coupled (jointly solvated) binding free energy isolates the JSI from the solvation-induced conformational effect.

In effect, the disjointly solvated coupling provides a more pertinent reference system compared to the typical reference system of the two solutes interacting in the gas phase. While the gas phase free energy may be perfectly reasonable in the case of rigid solutes, the binding properties of complex, highly flexible solutes, such as proteins, can be rather different in the gas phase compared to those in solution. In addition to facilitating strictly conformational effects, another benefit of the use of the disjointly solvated reference system is that it allows one to stabilize the relevant protonation states and counter ion structures of the proteins. Here, we provide the theoretical justification and analysis of this approach. The practical implementation of this generalized coupling scheme, which we refer to as partially connected molecular dynamics (PCMD), will be the subject of future work.

The article is organized as follows: first, we summarize the tools used in the analysis: the ME and its diagrammatic representation (Section 2), and the interaction expansion (Section 3). In Section 4, we define the JSI, present its physical interpretation, and propose a method for isolating its contribution to the binding free energy between two arbitrary solutes: PCMD. In Section 5, we show how the JSI reduces to the standard SII between rigid solutes, demonstrating that the JSI is a reasonable extension of the SII to flexible solutes. Finally, in Section 6, we compare the JSI with the more natural generalization of the SII, the indirect solvent-mediated interaction (i.e., the cavity interaction), and argue that the additional CE diagrams included in the JSI are critical for obtaining a more physically meaningful contribution of SII to the solute binding free energy. Additionally, Appendix A gives more detail on the connections between ME and the CE, and Appendix B discusses the convergence criteria for the JSI calculation.

## 2. The Mixture Expansion

We restrict the following discussion to pairwise potentials,
(1)$$V_{A,B}\left(R_{A},R_{B}\right)=\sum_{a\in A}\sum_{b\in B}u_{a,b}\left(R_{ab}\right),$$
for arbitrary site–site interactions, $u_{a,b}\left(R_{ab}\right)$, between site *a* belonging to species *A* and site *b* belonging to species *B*, which depend on the inter-site distance, $R_{a,b}$, and $R_{A}$ and $R_{B}$ correspond to the vector of the positions of all sites of species *A* and *B*, respectively.

Here, the term “species” can refer to a single type of molecule (for example all water molecules may constitute a single species), or it can refer to an arbitrary mixture (for example all molecules in a mixed solvent may be taken as a single species), or we may take a segment of a large protein to be a species. Note that, while a species will often correspond to a type of molecule, we may choose to assign two identical molecules (of the same type) to distinct species. For example, two methane molecules will be taken as distinct species when one computes the hydrophobic interaction between them. We will refer to the number of sites within a species, *A*, as $N_{A}$, and assume that this number appropriately accounts for the corresponding stoichiometry (for example, a species of water molecules, *W*, will have $N_{W}/3$ oxygen sites and $2N_{W}/3$ hydrogen sites). Note that, for both monatomic and molecular species, *A*, we will refer to the number of permutations of identical sites with factors of $1/N_{A}!$, which implicitly accounts for stoichiometry (e.g., $N_{W}!=N_{O}!\left(2N_{H}\right)!$ for water species, *W*). Sites can be taken to be atoms, in which case $u_{a,b}\left(R_{ab}\right)$ might be the sum of the Lennard-Jones + Coulomb interactions, or we can define distinct Lennard-Jones and Coulomb sites, and then force them to share the same position with a δ-bond (see Appendix A). In the interest of simplicity, we do not consider many-body interactions (e.g., electrostatic polarization).

The total free energy, *F*, of a system composed of a mixture of N species can be decomposed into the mixture expansion,
(2)$$F=F_{\mathrm{tot}}^{\left(1\right)}+F_{\mathrm{tot}}^{\left(2\right)}+F_{\mathrm{tot}}^{\left(3\right)}+\cdots +F_{\mathrm{tot}}^{\left(N\right)},$$
where $F_{\mathrm{tot}}^{\left(1\right)}=\sum_{n=1}^{N}F_{n}^{\left(1\right)}$ corresponds to the sum of the free energies of the N species in isolation. $F_{\mathrm{tot}}^{\left(2\right)}=\sum_{n=1}^{N}\sum_{m>n}^{N}F_{n,m}^{\left(2\right)}$ refers to the total second-order mixing free energy, i.e., the sum of mixing free energies of each pair of species in isolation, i.e., $F_{n,m}^{\left(2\right)}=F_{n⊕m}-F_{n}-F_{m}$ for a given pair of species *n* and *m*, with $F_{n⊕m}$ being the total free energy of the fully coupled system composed of species *n* and *m*. Higher-order terms are defined similarly, so that Equation (Equation 2) equals the total free energy of the fully coupled N-species system. In the interest of simplicity, we make use of unspecified constant volume mixing processes for each term in the ME (and in the interaction expansion below). However, the methods are general enough to be applied to any of the variety of constant volume mixing schemes [28], or to constant pressure processes (by multiplying the Boltzmann weight by $e^{-\beta pV}$ and integrating over V), etc.

The ME is most naturally represented through mixture diagrams, which are coarse-grained CE diagrams. Most mixture diagrams here will be composed of a collection of *E*-double-circles connected with *E*-double-lines. Each *E*-double-circle corresponds to the entirety of a single species, such that all sites within that species are fully coupled to one another (from the perspective of the CE, each site within the species is connected to each other site with an *e*-bond, hence the label “*E*-double-circle”, as described in Appendix A). For example the (canonical) configurational integral for species *A* is represented as the following 1-diagram: (3)
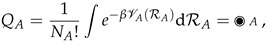

where $\beta =1/k_{B}T$, with $k_{B}$ being the Boltzmann constant, and *T* being the temperature; $V_{A}\left(R_{A}\right)$ is the full intra-species potential between each pair of sites within *A* (i.e., $V_{A}\left(R_{A}\right)=V_{A,A}\left(R_{A},R_{A}\right)$ with $u_{a^{\prime },a^{\prime \prime }}=0$ for $a^{\prime }=a^{\prime \prime }$), and the integration bounds over the $3N_{A}$-dimensional volume are implicit. Note that we use the term “*n*-diagram” to refer to a diagram with *n* double-circles.

The configurational integral for a mixture of species *A* and *B* can be represented as the following 2-diagram: (4)
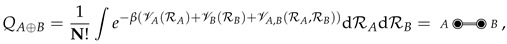

with $N!=N_{A}!N_{B}!$. Notice that the *E*-double-circles for *A* and *B* correspond to $\exp(-\beta V_{A})$ and $\exp(-\beta V_{B})$, respectively, and the *E*-double-line connecting them corresponds to $\exp(-\beta V_{A,B})$.

The (second-order) mixing free energy between *A* and *B* can be expressed as
(5)
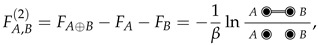

which corresponds to the change in the total free energy due to coupling each site in *A* to each site in *B* relative to the sum of the free energies of *A* and *B* independently (in isolation). Note that disconnected diagrams factor into connected subdiagrams, so the fraction in Equation (Equation 5) can be seen either as the connected 2-diagram divided by the disconnected 2-diagram, or as the connected 2-diagram divided by its two substituent 1-diagrams.

Similarly, the third-order mixing free energy between *A*, *B*, and *C* can be expressed as
(6)
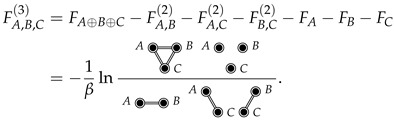

Note that each diagram in Equation (Equation 6) is a *complete* subdiagram of the $F_{A⊕B⊕C}$ diagram, with each double-circle connected to each other double-circle (taking the fully disconnected 3-diagram in the numerator to be three separate 1-diagrams). In general, excess mixing free energies in the mixture expansion are defined similarly in terms of complete subdiagrams.

In the next section, we will be interested in the free energies of systems on which we have imposed internal constraints. In particular, we will constrain the positions of two or more solutes, and then inquire about the dependence of the total system free energy as a function of their relative position. To do this, we will define a collective variable (CV), $s(R_{A},R_{B})$, which might be, for example, the center of mass distance between species *A* and *B*, and then insert a Dirac delta function into the integral which forces the CV to take on a given value, $R_{AB}$: (7)
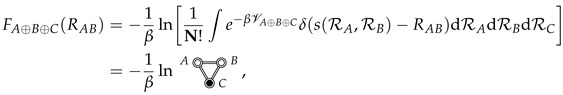

where we have whitened the *A* and *BE*-double-circles, indicating that the resulting integral has some dependence on a CV involving their respective coordinates, and we have abbreviated $V_{A⊕B⊕C}=V_{A}+V_{B}+V_{C}+V_{A,B}+V_{A,C}+V_{B,C}$, and $N!=N_{A}!N_{B}!N_{C}!$. Note that we have generalized the standard usage of white circles from the microscopic CE, taking advantage of the fact that we are sampling from MD simulations rather than solving integral equations.

Importantly, any diagram in which there is no path between white double-circles connected by a CV will be constant [23]. In particular,
(8)
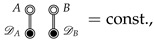

where $D_{A}$ and $D_{B}$ are arbitrary subdiagrams.

In addition to *E*-double-circles and *E*-double-lines, we will make use of δ-double-circles and δ-double-lines. Here, the δ-double-circle will refer to a species of rigid molecules, where the molecules are forced into the rigid geometry by Dirac delta functions: (9)
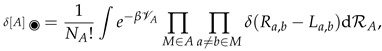

where the first product is indexed by molecule *M* of species *A*, *a* and *b* are distinct sites in molecule *M*, $L_{a,b}$ is the rigid distance between *a* and *b*, and $V_{A}$ is the total intra-species potential within *A* (the rigid molecules will still interact with each other). Note that, when the species is composed of a single rigid molecule, *M*, the integral corresponding to the δ-double-circle, δ[M], will collapse into an integral over the center of mass, $r_{M}$, and Euler angles (orientation), $\Omega_{M}$, of *M*: (10)
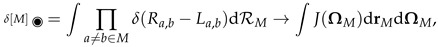

where *J*($\Omega_{M}$) is the Jacobian determinant for the coordinate transformation.

Finally, we introduce the δ-double-line which is used to partition the interactions due a single species, with the double-circle labeled *A*, into two (or more) double-circles, $A^{\prime }$ and $A^{\prime \prime }$: (11)
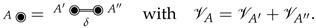

$A^{\prime }$ and $A^{\prime \prime }$ should be of the same dimensionality as *A* (with $N_{A}=N_{A^{\prime }}=N_{A^{\prime \prime }}$), and must have an identical distribution of site types (for example if *A* is a bulk water species, then $A^{\prime }$ and $A^{\prime \prime }$ must have the same number of oxygen and hydrogen sites). In terms of the CE, connecting the $A^{\prime }$ and $A^{\prime \prime }$ double-circles with a δ-double-line corresponds to connecting each site in $A^{\prime }$ to exactly one (identical) site in $A^{\prime \prime }$ via a δ-bond (e.g., hydrogen to hydrogen or oxygen to oxygen), forcing the pair to have precisely the same instantaneous position. Note that, due to indistinguishability between identical sites in *A*, the δ-double-line includes an extra factor of $N_{A}!$ (see Appendix A). Additionally, the components of the total system potential energy involving *A* must be partitioned between $A^{\prime }$ and $A^{\prime \prime }$ for Equation (Equation 11) to be valid. Writing Equation (Equation 11) explicitly, we have
$$\frac{1}{N_{A}!}\int e^{-\beta V_{A}}dR_{A}=\frac{N_{A}!}{{\left(N_{A}!\right)}^{2}}\int e^{-\beta V_{A^{\prime }}}e^{-\beta V_{A^{\prime \prime }}}\prod_{a\in A}\delta (|r_{a_{A^{\prime }}}-r_{a_{A^{\prime \prime }}}\left|\right)dR_{A^{\prime }}dR_{A^{\prime \prime }}$$
where $r_{a_{A^{\prime }}}$ and $r_{a_{A^{\prime \prime }}}$ correspond to the position vectors of site *a* belonging to double-circles $A^{\prime }$ and $A^{\prime \prime }$, respectively, and $V_{A}=V_{A^{\prime }}+V_{A^{\prime \prime }}$. Note that we only need a single product over the sites because the index *a* pairs each site in $A^{\prime }$ with exactly one site in $A^{\prime \prime }$.

One use of the δ-double-line is to separate the van der Waals and Coulomb interactions for one species into two double-circles (see the explanation of “alchemical intermediates” in Ref. [19]). Here, we use the δ-double-line to “duplicate” the solvent species in order to extract the disjointly solvated interaction from the total free energy in Section 6. Specifically, we will make use of
(12)
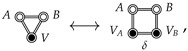

in which the introduction of the δ-double-line corresponds to the transformation
$$\int \cdots dR_{V}\to \;\int \int \;\cdots \prod_{v\in V}\delta (|r_{v_{A}}-r_{v_{B}}\left|\right)dR_{V_{A}}dR_{V_{B}},$$
and to the specific solute–solvent potential partitioning $V_{A,V_{A}}=V_{A,V}$ and $V_{A,V_{B}}=0$, and $V_{B,V_{B}}=V_{B,V}$ and $V_{A,V_{B}}=0$ (the partitioning of $V_{V}$ is inconsequential, so we leave it unspecified). The product of δ functions forces corresponding pairs of solvent sites, $v_{A}↔v_{B}$, to have the same instantaneous positions, so we may therefore (in general) partition the $V_{V}$, $V_{A,V}$, and $V_{B,V}$ potentials between $V_{A}$ and $V_{B}$ in whichever way we prefer, with each partitioning leading to a different ME diagram correspondence. For example, partitioning the solute–solvent electrostatics (el) and van der Waals (vw) interactions leads to 

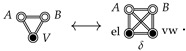


## 3. The Interaction Expansion

While the mixture expansion provides a helpful analysis tool on its own (see Ref. [19]), the analysis of partially connected mixture diagrams requires the use of the interaction expansion of the inter-species potential distribution cumulant generating function [29,30]. While the mixture expansion can be seen as the cluster expansion of double-circles, the interaction expansion is the cluster expansion of double-lines.

The interaction expansion of a given *n*-order mixing free energy, $F_{A,B,\cdots ,Z}^{\left(n\right)}$ (corresponding to a complete *n*-diagram), is the sum of cluster cumulant functions (CCFs), *K*, due to the connected (but not necessarily complete) subdiagrams. For example, the (excess) third-order mixing free energy (Equation (Equation 6)) can be expanded as
(13)
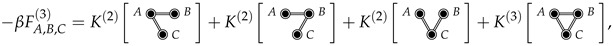

where, for example, the third-order CCF is equal to
(14)
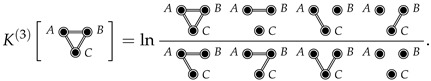

Notice that $K^{\left(3\right)}$ is a third-order CCF given that it contains three double-lines (the fact that it contains three double-circles is coincidental). The interaction expansion of an *n*-order mixing free energy will include CCFs from order n−1 (corresponding to the spanning trees) up to order C2n=n!/(n−2)!2! (corresponding to the fully connected CCF).

We will also need interaction expansions of arbitrary diagrams (not just excess mixing free energies), for example,
(15)
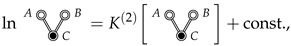

where we have used the fact that diagrams with no path between the white circles is constant (see Equation (Equation 8)).

Importantly, CCFs of diagrams with δ-double-lines can be simplified in the following way (using the analog of Equations (Equation 12) and (Equation 14)): (16)
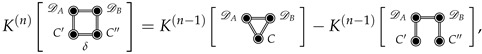

where $D_{A}$ and $D_{B}$ are arbitrary subdiagrams, and all double-lines except for the δ-double-line correspond to arbitrary double-line connectivities.

## 4. The Joint Solvation Interaction

The total SII between species *A* and *B* is
(17)
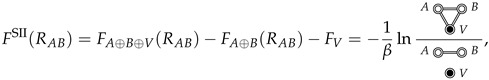

where $F_{A⊕B⊕V}\left(R_{AB}\right)$ corresponds to the free energy of the fully solvated *A* and *B* (with solvent *V*), fixed at a distance $R_{AB}$, $F_{A⊕B}\left(R_{AB}\right)$ corresponds to the free energy of *A* and *B* in the gas phase, fixed at the same distance, and $F_{V}$ is the free energy of the pure solvent, which has no $R_{AB}$ dependence (it is constant).

When A=a and B=b are spherical particles with isotropic interactions, $F_{a⊕b}\left(R_{ab}\right)$ reduces to the pair interaction, $u_{a,b}\left(R_{ab}\right)$, and $F_{a,b}^{\mathrm{SII}}\left(R_{ab}\right)$ corresponds to the standard definition of the SII (plus a constant):(18)$${\left[F^{\mathrm{SII}}\left(R_{ab}\right)\right]}_{\mathrm{sph}.}+\mathrm{const}.=-\frac{1}{\beta }\ln g\left(R_{ab}\right)-u_{a,b}\left(R_{ab}\right)\equiv -\frac{1}{\beta }\ln y\left(R_{ab}\right),$$
where $g(R_{ab})$ is the radial distribution function for *a* and *b* dissolved in solvent *V*, and $y(R_{ab})$ is the “indirect” correlation function [14,15,31] between spherical *a* and *b* (we generalize y(R) in Section 6).

When *A* and *B* are rigid solutes, $F_{A⊕B}\left(R_{AB}\right)$ has an energetic component corresponding to the (gas phase) orientation-averaged interaction energy at fixed $R_{AB}$, ${\langle V_{A,B}\left(R_{AB}\right)\rangle }_{\mathrm{gas}}$, as well as an entropic component due to the nonuniform orientational distribution of *A* and *B* in the gas phase which is nonzero since some relative orientations may be more likely than others due to nonisotropic interactions (making it repulsive compared to the noninteracting limit, $R_{AB}\to \infty$).

In the case of flexible *A* and *B*, however, while Equation (Equation 17) is well defined, it is not particularly helpful. For example, if either *A* or *B* are large proteins that unfold/refold outside of aqueous solvent, then their conformations in the gas phase might not be relevant to their binding properties in solution. The comparison between the illustrations in panels a and b of Figure 2 depicts a hypothetical example of the difference in the gas phase and the solution phase conformations of interacting flexible solutes. This kind of incompatibility provides the motivation for the JSI: we need a more relevant reference to compare the solvated free energy to (i.e., we need a better denominator in an expression analogous to Equation (Equation 17)).

With this goal in mind, consider the total free energy of independently solvated species *A* and *B*, which is the sum of their respective free energies in independent solvents: (19)
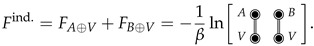

Clearly, $F^{\mathrm{ind}.}$ can be interpreted as the total free energy of two distinct systems: *A* and *B* dissolved in different solvent boxes. But one might alternatively interpret it as *A* and *B* dissolved in the same *V*, such that they are far enough away that they are independently solvated (e.g., with $R_{AB}\to \infty$). This is an acceptable interpretation in the thermodynamic limit of the number of solvent molecules, $N_{V}\to \infty$ (so we may freely double-count *V* without issue), and, in practice, this is acceptable if certain convergence conditions are met (see Appendix B).

Following a similar reasoning, we define the *disjointly solvated interaction* between *A* and *B*,
(20)
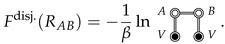

This type of coupling is a bit unconventional, so let us consider it carefully. The first issue is that we have duplicated the solvent again, which we justify in Appendix B (briefly: if $N_{V}$ is large enough, spurious contributions due to the duplicated solvent will sum to a constant). The second issue is that we are now considering a partially connected diagram. This means that each solute is fully coupled to its own solvent, while also being fully coupled to the other solute, while neither solute is coupled to the other solvent. Such a scenario is entirely unphysical! Notice that we make no attempt at the moment to express $F^{\mathrm{disj}.}$ as a sum of free energies as we did in the prior formulas. Figure 2d shows an illustration of how one might imagine this kind of coupling.

Issues aside, let us appreciate what we have gained from this definition. Notice that both *A* and *B* are fully solvated so the correct conformations and protonation states may be stabilized. At the same time, *A* and *B* may interact with each other, unencumbered by nonlinear joint solvation effects. For example, as *A* comes into contact with *B*, the solvent molecules coupled to *B* do not need to rearrange due to excluded volume effects with *A*. Of course, *A* will interact *indirectly* with the *B* solvent, but these effects are strictly mediated through $R_{AB}$-dependent conformational changes of *B*. In the case where *A* and *B* are large proteins, $F^{\mathrm{disj}.}$ is clearly a better reference than the gas phase *A*-*B* free energy for analyzing SII in a meaningful way. Therefore, we define the JSI as follows: (21)
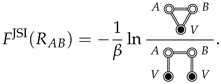

This definition of the JSI isolates the effect on the total free energy due to the $R_{AB}$-dependent nonlinear joint solvation of *A* and *B*. Collective effects such as the mutual excluded volume between *A*, *B*, and the surrounding solvent are fully captured, and thus the conventional picture of the hydrophobic interaction is recovered. Additionally, effects due to the simultaneous interaction of single solvent molecules with both *A* and *B* are also isolated here, and the conventional picture of the hydrophilic interaction (for example with water molecules forming hydrogen bond bridges between two solutes) and other related effects are recovered as well [14].

In order to compute such a quantity, one needs to make use of PCMD. Taking inspiration from thermodynamic integration [32,33,34,35], free energy perturbation [34,35,36,37], and the general use of alchemical intermediates [19], PCMD introduces coupling parameters which turn on/off interactions between specific pairs of species rather than turning on/off all interactions between one species and all other species.

For example, in calculating the JSI between two molecules, *A* and *B*, in water, we would place *A* and *B* within the same simulation box so that the CV, $s(R_{A},R_{B})$, is well defined. We also place two copies of the water species within the same box (labeling them $W_{A}$ and $W_{B}$). We turn on the potentials $V_{A,B}$, $V_{A,W_{A}}$, and $V_{B,W_{B}}$, but set all other potentials to 0 ($V_{A,W_{B}}=V_{B,W_{A}}=V_{W_{A},W_{B}}=0$). This kind of coupling is represented pictorially by the diagram in Equation (Equation 20) and the illustration in Figure 2d. Note that, in Figure 2d, we depict the two solvent species and the interacting solutes separately for visual clarity. However in the calculation described here, all four species (*A*, *B*, $W_{A}$, and $W_{B}$) all occupy the same simulation box, so all interactions are well defined. Note that, with $V_{W_{A},W_{B}}=0$, the two solvent species can exist within the same volume without issue.

Importantly, since $V_{A,W_{B}}=0$, the sites belonging to $W_{B}$ may freely overlap with *A*, and similarly between $W_{A}$ and *B*, and between $W_{A}$ and $W_{B}$, thus avoiding joint solvation effects. Since *A* and *B* are fully solvated (by $W_{A}$ and $W_{B}$, respectively), they do not unfold/refold as they would in the gas phase. At the same time, *A* and *B* interact directly, leading to the disjointly solvated interaction. The total forces due to this coupling need to be calculated for each time step in a PCMD simulation. Since the potential energy is pairwise additive (Equation (Equation 1)), we can calculate the forces due to each nonzero potential independently, and then simply add them together in order to propagate the dynamics for the disjointly solvated coupling.

Using this setup, we can calculate the free energy profile between *A* and *B* in this partially connected potential, giving $F^{\mathrm{disj}.}\left(R_{AB}\right)$. Finally, the free energy profile between *A* and *B* dissolved in the same water box is computed, giving $F_{A⊕B⊕V}\left(R_{AB}\right)$, and the JSI is computed from the difference, $F^{\mathrm{JSI}}\left(R_{AB}\right)=F_{A⊕B⊕V}\left(R_{AB}\right)-F^{\mathrm{disj}.}\left(R_{AB}\right)$.

## 5. Equivalence of the Joint Solvation Interaction with the Solvent-Induced Interaction for Rigid Solutes

Now we demonstrate the equivalence of our definition of the JSI (Equation (Equation 21)) with the standard definitions of the SII between spherical particles with isotropic interactions dissolved in solvent *V* and the SII between rigid molecules dissolved in *V*.

In the case of spherical particles, A=a and B=b, we replace the solute double-circles with single-circles, and the *E* double-lines connecting them with *e*-bonds (see Appendix A),
(22)
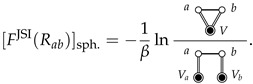

To demonstrate the equivalence with the spherical SII, it suffices to show that the log of the partially connected diagram in the denominator in Equation (Equation 22) equals the pair interaction $-\beta u_{a,b}$ (plus a constant). This simply follows from the fact that both *a* and *b* are articulation circles [22,23] (i.e., removing either splits the diagram into disconnected subdiagrams in which at least one subdiagram has no white circles). Therefore, we can first choose the position of spherical particle *a*, $r_{a}$, as the origin, change the coordinates in the $V_{a}$ integral to be relative to $r_{a}$, and then factor out the entire $V_{a}$ integral (since it is constant with respect to the position of the origin in the absence of external potentials). We can do the same with the $V_{b}$ integral. Now notice that the factored out $V_{a}$ and $V_{b}$ integrals have no $R_{ab}$ dependence, and the log of the diagram reduces to
(23)
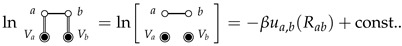


In order to make a similar reduction for rigid (but non-spherical) solutes *A* and *B*, we now sketch out the analogous procedure in more detail. Consider diagrams with the following structure: 
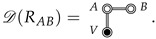

We want to rearrange the corresponding integral as follows:(24)$$\begin{array}{cc} D(R_{AB})\; & =\int dR_{A}e^{-\beta V_{A}}\int dR_{V}e^{-\beta (V_{V}+V_{A,V})}\int dR_{B}\delta \left(s\left(R_{A},R_{B}\right)-R_{AB}\right)e^{-\beta (V_{B}+V_{A,B})}\\ \; & =\int dR_{A}e^{-\beta V_{A}}{\tilde{Q}}_{V}\left(R_{A}\right){\tilde{Q}}_{B}\left(R_{A},R_{AB}\right), \end{array}$$
where $s(R_{A},R_{B})$ is the transformation which takes the configurations of *A* and *B* to the chosen CV (for example the center of mass distance), and the delta function, δ, filters out configurations in the $R_{B}$ integrals with $s\left(R_{A},R_{B}\right)\neq R_{AB}$. Note the absence of $V_{V,B}$ in Equation (Equation 24) since *V* and *B* are disconnected in the diagram $D(R_{AB})$. Also, note that ${\tilde{Q}}_{V}\left(R_{A}\right)$ and ${\tilde{Q}}_{B}\left(R_{A},R_{AB}\right)$ result from carrying out the $R_{V}$ and $R_{B}$ integrals, respectively, *inside* the $R_{A}$ integral, so that both still have dependence on the *A* configuration, $R_{A}$.

In the case where *A* is a rigid molecule, the remaining Boltzmann factor, $\exp[-\beta V_{A}]$, is replaced with a product of delta functions restraining all *A* inter-atomic distances into their rigid conformation (the *AE*-double-circle is replaced with a δ-double-circle), and the 3$N_{A}$-dimensional integral reduces to a six-dimensional integral over the center of mass position, $r_{A}$, and the Euler angles, $\Omega_{A}$ (see Equation (Equation 10)):(25)
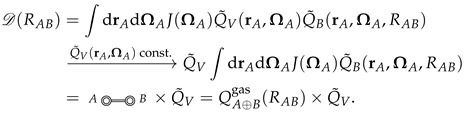

Noting that $r_{A}$ and $\Omega_{A}$ can be taken as the origin and orientation of the coordinate system in the *V* integral, respectively, we see that ${\tilde{Q}}_{V}\left(r_{A},\Omega_{A}\right)$ is constant in the absence of external potentials (i.e., space is homogeneous and isotropic), and thus factors out (in the second line of Equation (Equation 25)). As a result we are left with the gas phase A⊕B configurational integral, $Q_{A⊕B}^{\mathrm{gas}}\left(R_{AB}\right)$, multiplied by the constant ${\tilde{Q}}_{V}$, which corresponds to the configurational integral of solvent *V*, solvating the rigid *A* (independently of *B*). Continuing this argument, we find that
(26)
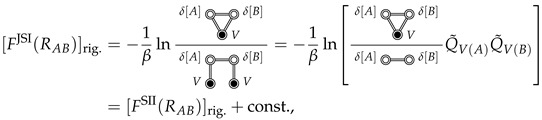

where ${\tilde{Q}}_{V(A)}$ and ${\tilde{Q}}_{V(B)}$ are the (constant) configurational integrals for solvent *V*, solvating rigid *A* and rigid *B*, respectively.

## 6. Comparison with the Cavity Interaction

Ben Naim defines the hydrophobic interaction between rigid solutes to be the indirect solvent-mediated interaction, i.e., the cavity interaction, $F^{\mathrm{cav}.}$, which is defined in terms of the indirect correlation function, y(R) [14,15,31]. When the solutes are spherical particles, y(R) is defined by Equation (Equation 18). For flexible solutes, one could generalize the definition of y(R) as a sum of certain indirect CE diagrams [15], which is conveniently summarized by the following ME diagram: (27)
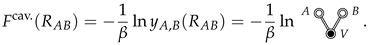


As a side note, Pratt and Chandler extended the definition of the hydrophobic interaction to describe the (flexible) aggregation of spherical solutes [15]. However, the simplest adaptation of their definition to the context of flexible solutes binding together is simply the total SII. This is clear from the definition that they use (due to Ben Naim),
(28)$$\delta F^{\mathrm{HI}}=\Delta \mu_{M}-\sum_{n\in M}\Delta \mu_{n}=\langle F^{\mathrm{SII}}\rangle ,$$
where $\Delta \mu_{M}$ is the excess chemical potential (with respect to the ideal gas) of “molecule” *M* (i.e., the aggregation), and $\Delta \mu_{n}$ is the excess chemical potential of the nth (spherical) particle in *M*, and $\langle F^{\mathrm{SII}}\rangle$ is the ensemble average SII between all particles aggregated into *M* (dissolved in a given solvent).

Returning to the comparison between the JSI (Equation (Equation 21)) and $F^{\mathrm{cav}.}$ (Equation (Equation 27)), we first point out that computing the diagram in the right-most expression in Equation (Equation 27) only requires turning off the *A*-*B* interaction in a PCMD free energy calculation. This gives the free energy of the *A* and *B* cavities dissolved in the solvent, *V*, as a function of $R_{AB}$. Since cavities interact with the solvent, this will include the indirect solvent-mediated contribution to the free energy. However, since cavities do not interact with other cavities (see Equations (Equation 18) and (Equation 27)), the region of configuration space contributing to $F^{\mathrm{cav}.}$ will include unphysical geometries in which *A* and *B* overlap. Therefore, while $F^{\mathrm{cav}.}$ may prove to be a useful component of the total SII, we argue that the JSI is of more immediate utility in understanding the binding of *A* with *B* since $F^{\mathrm{JSI}}$ will not include contributions from these types of overlapping *A*-*B* configurations. Figure 2c shows an illustration of how one might imagine this kind of coupling, with *A* and *B* dissolved in the same water box, but decoupled from each other (they may overlap freely).

Finally, we explicitly compute the difference between $F^{\mathrm{cav}.}$ and $F^{\mathrm{JSI}}$. To do this, we employ the following trick: starting with Equation (Equation 21), we duplicate the *V E*-double-circle of $N_{V}$ solvent sites (with $N_{V}$ sufficiently large) in the numerator, connect the resulting two *V E*-double-circles with a δ-double-line, and partition the solute–solvent potential between the two solvent species in the same way as Equation (Equation 12): (29)
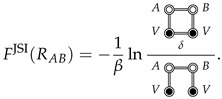

In other words, we duplicate the solvent (in the same spirit as the disjointly solvated interaction diagram), but pair up each site in one double-circle with an equivalent site in the other double-circle, and force each pair to have the exact same instantaneous position via the δ-double-line (see Appendix A). From this perspective, one might see $F^{\mathrm{JSI}}$ as the free energy cost of forcing the two solvent species to have identical instantaneous configurations within the ensemble average.

Now that the numerator and denominator have the same dimensionality, we expand each diagram into interaction distribution CCFs (see Section 3), and simplify, resulting in
(30)
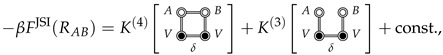

where we use Equation (Equation 16), which leads to the cancellation of terms such as

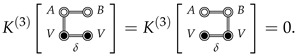

Notably, the constant in Equation (Equation 30) is simply $-\ln Q_{V}$, which serves as an inconsequential reminder that we divided a 3-diagram by a 4-diagram in our definition of $F^{\mathrm{JSI}}$. After expanding out the CCFs in Equation (Equation 30), and simplifying with Equation (Equation 16), we obtain
(31)
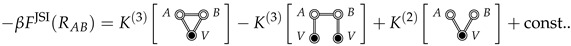

We will refer to the three CCFs on the right side as $K^{\left(3\right)}\left[D_{3}\right]$, $K^{\left(3\right)}\left[D_{4}\right]$, and $K^{\left(2\right)}\left[D_{3}\right]$ (from left to right). In light of Equation (Equation 15), it is clear that $F^{\mathrm{JSI}}$ contains the $R_{AB}$-dependent part of $F^{\mathrm{cav}.}$ (which is equal to the second-order CCF, $K^{\left(2\right)}\left[D_{3}\right]$), making it an acceptable candidate for a generalization of the SII. However, $F^{\mathrm{JSI}}$ surpasses $F^{\mathrm{cav}.}$ in that it also contains the third-order CCF, $K^{\left(3\right)}\left[D_{3}\right]$, which corresponds to the excess free energy contribution due to *simultaneously* coupling *A*, *B*, and *V* with each other (see Equation (Equation 14)). Therefore, $K^{\left(3\right)}\left[D_{3}\right]$ includes the CE diagrams which cancel contributions from the unphysical *A*-*B* conformations that are included in $F^{\mathrm{cav}.}$. The $K^{\left(3\right)}\left[D_{4}\right]$ term looks a bit out of place, since we are subtracting a 4-diagram CCF from 3-diagram CCFs; however, in the limit of $N_{V}\to \infty$, transformation into the *f*-bond representation (see Appendix A) shows that the entirety of $K^{\left(3\right)}\left[D_{4}\right]$ is contained in the expansion of $K^{\left(3\right)}\left[D_{3}\right]$, so subtracting it from $K^{\left(3\right)}\left[D_{3}\right]$ is simply removing the $F^{\mathrm{disj}.}$ contribution from the SII. However, in a calculation with finite $N_{V}$, there will be spurious contributions from higher-order *f*-bond CE diagrams, which we address in Appendix B.

In order to contextualize this comparison, we perform a similar interaction expansion analysis on the total SII. We have asserted that the total SII is the sum of the JSI and the solvation-induced conformational effect. To see this, we rewrite the SII (Equation (Equation 17)) as
(32)
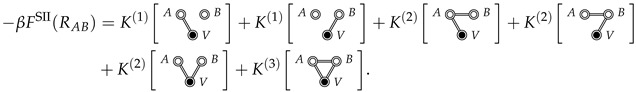

Rearranging Equation (Equation 31) and inserting it into Equation (Equation 32), we find
(33)
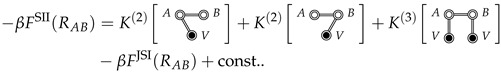

We can identify the three CCFs shown in Equation (Equation 33) as the $R_{AB}$-dependent solvation-induced conformational effect. In particular, the two $K^{\left(2\right)}$ CCFs represent the conformational effects due to solvating *A* and *B* individually, and $K^{\left(3\right)}\left[D_{3}\right]$ represents the excess contribution due to disjointly solvating *A* and *B* simultaneously. The sum of these three CCFs is equal to the $R_{AB}$-dependent part of $-\beta (F^{\mathrm{disj}.}-\langle V_{AB}\rangle )$, as expected from comparing Equations (Equation 17) and (Equation 21).

## 7. Conclusions

The net interaction between two solutes in solution includes the direct interaction potential between the solutes, as well as indirect effects mediated by the surrounding solvent. These indirect interactions include the effect of the solvated conformational distribution (for example the ensemble of relevant protein folds within a given set of thermodynamic conditions), and also the effects due to the simultaneous accommodation of the two solutes within the solvent, the latter of which we refer to as the joint solvation interaction. Joint solvation effects are well appreciated in the literature discussing the hydrophobic interaction, and more generally on solvent-induced interactions. However the typical definitions assigned to these effects are intended to describe the solvation of rigid solutes, and are (arguably) unsuited for complex, highly flexible solutes. This study is our attempt to re-frame the idea of solvent-induced interactions towards the specific effects due to joint solvation.

To do this, we have extended our mixture expansion free energy decomposition formalism to include partially connected couplings by using the interaction expansion of the inter-species potential distribution cumulant generating function. The finer detail allowed by the resulting decomposition scheme is due to the consideration of unconventional couplings which probe specific types of collective effects. Here, we find that the thermodynamics of joint solvation are very naturally expressed in this language.

We have presented a definition of the joint solvation interaction within the framework of the interaction expansion, which we argue offers the most physically meaningful components of the free energy specifically describing joint solvation. We give a physically motivated justification of our definition, we show that this definition reduces to the typical definition of the solvent-induced interaction in the case of rigid solutes, and we explain the benefits of this definition compared to the cavity interaction, which is a more natural extension of earlier ideas, but includes contributions from effects which are unphysical (since it considers configurations in which the solutes overlap, for example).

Our definition requires the use of unphysical (partially connected) couplings, and an unphysical system setup (with a duplicated solvent species), but an analysis of the corresponding microscopic cluster expansion (see the Appendix A and Appendix B) demonstrates that these quirks are well justified, and lead to a quantity that is (at least) reasonably physical (up to unimportant constant terms). However, everything presented here is still theoretical, and a practical implementation in molecular dynamics software (i.e., partially connected molecular dynamics) does not exist at the moment. Such an implementation requires a more generalized inter-atomic force evaluation, which may involve fairly invasive changes to existing codes. Nonetheless, we believe that the possibility of highly specific free energy decomposition studies—including (but not limited to) analyses of joint solvation interactions between flexible solutes—offers a unique benefit towards our understanding of complicated solvation phenomena, such as self-assembly, protein aggregation, and protein folding. Therefore, such an implementation effort will be the subject of future work.

## Figures and Tables

**Figure 1 entropy-26-00749-f001:**
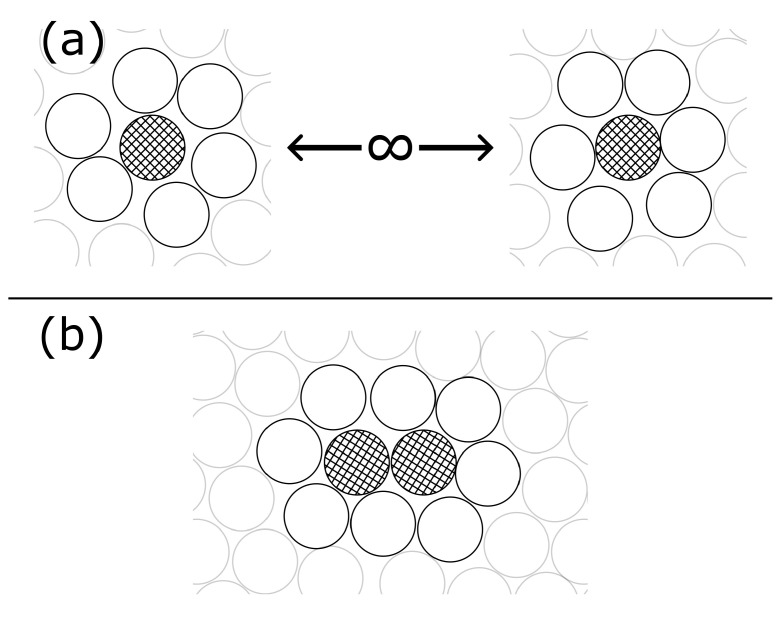
Illustration depicting two solute particles (cross-hatched) and their nearest neighbor solvent particles. Panel (**a**) shows disjointly solvated solutes and panel (**b**) shows jointly solvated solutes.

**Figure 2 entropy-26-00749-f002:**
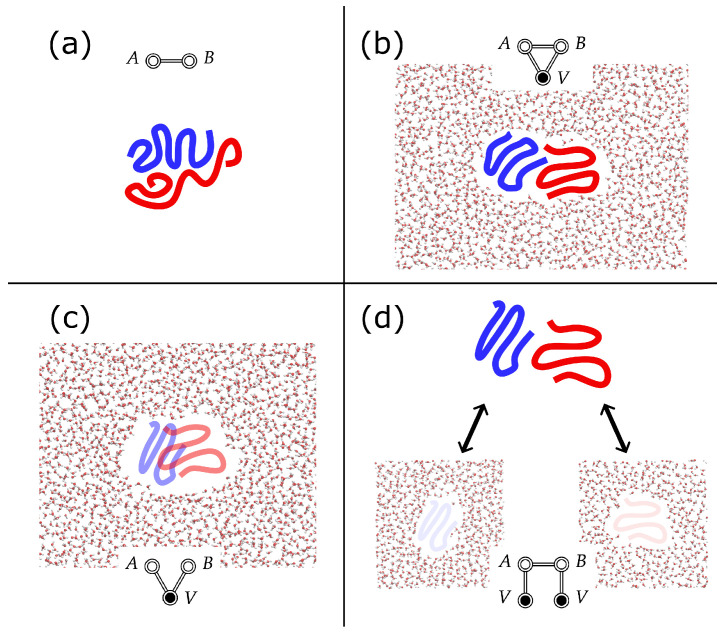
Illustration depicting four types of couplings between flexible molecules, *A* (blue) and *B* (red): (**a**) gas phase *A*–*B* in which the configurations adopted by each molecule are not necessarily related to their solvated configurations, (**b**) fully solvated, fully coupled *A*–*B* (i.e., the standard coupling), (**c**) cavity *A*-*B* coupling in which both molecules are fully coupled to the same solvent, but are decoupled from each other, allowing for unphysical configurations in which *A* and *B* overlap, (**d**) disjointly solvated *A*-*B* in which each molecule is solvated in its own solvent while *A* and *B* are fully coupled (arrows represent the solute-solvent couplings). Note that, in panel (**d**), both molecules and both solvent species occupy the same simulation box so that the distance $R_{ab}$ is well defined.

## Data Availability

No new data were created or analyzed in this study. Data sharing is not applicable to this article.

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
