# Peer review of "The Joint Solvation Interaction"

_entropy, 2024, doi:10.3390/e26090749_

Round 1

Reviewer 1 Report

Comments and Suggestions for Authors

In the manuscript entitled “The Joint Solvation Interaction”, A. Hassanali and C. K. Egan have used the mixture expansion(ME), an coarse-grained version of the cluster expansion to “bundle” of molecules, to separate joint solvation interaction(JSI) from the total solvent induced interactions(SII). Through ME, the joint solvation interaction has been defined by the free energy difference between a state with full interactions among all solutes and solvent and disjointly-solvated state, in which each solute is interacting with their own solvent, given the interaction between solutes. Then, the authors showed that this new definition becomes equivalent to the total SII when the solutes are rigid. Also, by employing interaction expansion, they showed that the typical hydrophobic, or cavity interaction is naturally included in their JSI, generalizing such indirect solvent-mediated interaction.

The manuscript was very interesting to me during I’ve read this, and it could do also for the readers in Entropy, as well as the researchers in the field of macromolecular science, such as polymeric materials or biophysical chemistry. From the theoretical point of view, therefore, I recommend this manuscript to publish in the journal Entropy. But before final acceptance, I comment some minor points to the author, hoping they assign these issues:

1.     For the mixture expansion and interaction expansion approaches for JSI, I can agree well to them. But minorly, could the authors note about the remaining cluster terms from total SII, and show whether it corresponds to fully interactions due to the conformational change or includes them?

2.     For partially-coupled MD: The authors proposed MD scheme to compute JSI. This scheme includes some calculations of unphysical, jointly solvated state, by doing two calculations—one is with turning on the A-B and A-V interaction and turning off B-V one, and the other is with turning on the A-B and B-V interaction and turning off A-V one. In such calculations, could there be some technical issues related to the inconsistency of system size? Also, I’m wondering that such calculation represents the 3-diagram with single V double circle connecting with on of A or B, not the 4-diagram with V-A-B-V connection.

3.     Could the authors explain the equation (11) explicitly?

4.     Page 4, line 141: I think the term “ideal mixture” is not proper for the denominator of eqn (5), because ideal mixture means that all A-A, B-B, A-B interactions are same, not that A-B interactions are turned off!!

5.     The remains are some typos:

Page 3, line 97: R_A and R_A à R_A and R_B

Page 4, line 131: v_a,b à u_a,b

Author Response

We are indebted to the reviewer for the extremely helpful feedback. Here are our responses:

Comment 1:  For the mixture expansion and interaction expansion approaches for JSI, I can agree well to them. But minorly, could the authors note about the remaining cluster terms from total SII, and show whether it corresponds to fully interactions due to the conformational change or includes them?

Response 1:  Excellent point! We have added this to the discussion in section 6. Thank you for the suggestion!

Comment 2:  For partially-coupled MD: The authors proposed MD scheme to compute JSI. This scheme includes some calculations of unphysical, jointly solvated state, by doing two calculations—one is with turning on the A-B and A-V interaction and turning off B-V one, and the other is with turning on the A-B and B-V interaction and turning off A-V one. In such calculations, could there be some technical issues related to the inconsistency of system size? Also, I’m wondering that such calculation represents the 3-diagram with single V double circle connecting with on of A or B, not the 4-diagram with V-A-B-V connection.

Response 2: Unfortunately the 4-diagram is necessary for this calculation. In the previous submission, our explanation of the PCMD calculation was not written clearly enough. We have rewritten this (at the end of section 4), so hopefully it will make sense now. The point is that A and B both need to be solvated and interacting with each other, but with effects due to the joint solvation removed. Therefore at each timestep we need the total force due to this coupling. The 3-diagrams described by the reviewer will only account for 2nd-order solvent-induced conformational effects, but not the 3rd-order effect (see the added discussion in section 6 from the previous point).

Comment 3:  Could the authors explain the equation (11) explicitly?

Response 3:  We thank the reviewer again for a very helpful suggestion. After working out the details, we realized that the previous explanation was hand-wavy and actually a bit dubious. We cleaned up the explanation and now show details at the end of section 2.

Comment 4:  Page 4, line 141: I think the term “ideal mixture” is not proper for the denominator of eqn (5), because ideal mixture means that all A-A, B-B, A-B interactions are same, not that A-B interactions are turned off!!

Response 4:  The reviewer is absolutely correct here. This has now been corrected.

Comment 5:  The remains are some typos: Page 3, line 97: R_A and R_A à R_A and R_B Page 4, line 131: v_a,b à u_a,b

Response 5:  We have corrected these typos.

Thank you again!

Reviewer 2 Report

Comments and Suggestions for Authors

Nice work. Exceptionally clearly presented and written.

Author Response

Thank you!

Reviewer 3 Report

Comments and Suggestions for Authors

In the manuscript, the authors present a theoretical framework for the calculation of solvent effect components that can be applied to flexible molecules in solution that interact with each other. The presented theoretical model can be applied in the study of a wide range of phenomena, including processes involving interactions between biomolecules in solution.

The authors define Joint Solvation Interaction, a term that quantifies solvent interactions for solvents that simultaneously accommodate large flexible and interacting solutes.

Also, the development of tunable partially connected molecular dynamics necessary for the application of this theoretical method is proposed. The development of this method could prove to be a special challenge and it remains to be seen how the proposed model helps in understanding solvation.

The manuscript is suitable for a special issue, it is well written, well structured and clearly presented, it also contains diagrams and intuitive representations for the benefit of the general reader, and therefore I recommend the publication of this paper.

If the authors consider it appropriate a note or brief explanation on how the interactions in Figure 2d are treated given that the interacting parts of A and B are simultaneously interacting with the solvent in separate boxes.

Would taking the simulation trajectories for A and B from Fig. 2b (standard coupling) separately and subsequently placing them separately in the respective solvent boxes, and then allowing the solvent to interact and adapt to these obtained trajectory structures also make sense for the JSI calculation ?

Minor point, misprint on page 12 line 383.

Author Response

Comment 1:   If the authors consider it appropriate a note or brief explanation on how the interactions in Figure 2d are treated given that the interacting parts of A and B are simultaneously interacting with the solvent in separate boxes.

Response 1:  We thank the reviewer for the suggestion. We have cleaned up our explanation of the PCMD calculation at the end of section 4. We have also clarified that Figure 2d visually depicts separate boxes for clarity, while in the real calculation, all species occupy the same simulation box. Since the two solvent species do not interact with each other, they can both occupy the same volume without issue.

Comment 2:  Would taking the simulation trajectories for A and B from Fig. 2b (standard coupling) separately and subsequently placing them separately in the respective solvent boxes, and then allowing the solvent to interact and adapt to these obtained trajectory structures also make sense for the JSI calculation ?

Response 2:  Unfortunately, the partially-connected forces need to be evaluated at each timestep of the simulation in order to get the partially-connected PMF. We are unsure whether we understand the suggestion exactly, but it might be possible to get an estimate of the PMF from perturbation theory using the suggested approach. Depending on how difficult it is to implement PCMD in a real MD code, this might be useful to get an estimate of the effect, but the authors aren’t confident enough about the details to mention this in the manuscript.

Comment 3:  Minor point, misprint on page 12 line 383.

Response 3:  Sorry, we should have asked for a clarification earlier. We aren’t sure what misprint the review is referring to. Could you please specify?

Thank you!